# Removal of V(V) From Solution Using a Silica-Supported Primary Amine Resin: Batch Studies, Experimental Analysis, and Mathematical Modeling

**DOI:** 10.3390/molecules25061448

**Published:** 2020-03-23

**Authors:** Xi Huang, Zhenxiong Ye, Lifeng Chen, Xujie Chen, Caocong Liu, Yuan Yin, Xinpeng Wang, Yuezhou Wei

**Affiliations:** 1Advanced Nuclear Energy Research Group, College of Physics and Optoelectronic Engineering, Shenzhen University, Shenzhen 518060, China; yinyuan@szu.edu.cn; 2College of Chemistry and Chemical Engineering, Guangxi University, Nanning 530004, China; 1715308005@st.gxu.edu.cn; 3School of Nuclear Science and Engineering, Shanghai Jiao Tong University, 800 Dong Chuan Road, Shanghai 200240, China; chenlf@sjtu.edu.cn (L.C.); yzwei@sjtu.edu.cn (Y.W.); 4Guangxi Key Laboratory of Processing for Non-Ferrous Metallic and Featured Materials, School of Resources, Environment and Materials, Guangxi University, Nanning 530004, China; 18894826954@163.com (X.C.); liucaocong@163.com (C.L.)

**Keywords:** pentavalent vanadium, removal, silica-supported resin, mathematical modeling

## Abstract

Every year, a large quantity of vanadium-containing wastewater is discharged from industrial factories, resulting in severe environmental problems. In particular, V(V) is recognized as a potentially hazardous contaminant due to its high mobility and toxicity, and it has received considerable attention. In this study, a silica-supported primary amine resin (SiPAR) was prepared by in-situ polymerization, and the V(V) adsorption from the solution was examined. The as-prepared resin exhibited fast adsorption kinetics, and it could attain an equilibrium within 90 min for the V(V) solution concentration of 100 mg/L at an optimum pH of 4, whereas the commercial D302 resin required a treatment time of more than 3 h under the same conditions. Furthermore, the maximum adsorption capacity of the resin under optimum conditions for V(V) was calculated to be 70.57 mg/g. In addition, the kinetics and isotherm data were satisfactorily elucidated with the pseudo-second-order kinetics and Redlich–Peterson models, respectively. The silica-based resin exhibited an excellent selectivity for V(V), and the removal efficiency exceeded 97% in the presence of competitive anions at 100 mmol/L concentrations. The film mass-transfer coefficient (k_f_) and V(V) pore diffusivity (D_p_) onto the resins were estimated by mathematical modeling. In summary, this study provided a potential adsorbent for the efficient removal of V(V) from wastewater.

## 1. Introduction

Vanadium is an important alloying element that is mainly used in the steel industry due to its excellent properties, including its high strength, toughness, and wear resistance [1,2]. In addition, vanadium chemicals are widely used in catalysts, ceramics, pigments, and batteries [3]. However, industrial wastewater containing soluble vanadium is inevitably generated in these processes. For example, large amounts of vanadium-bearing steel slag are produced after steelmaking, in which vanadium is leached with H_2_SO_4_ and then precipitated with ammonium salt for reuse [4]. The residual filtrate containing a low concentration of vanadium becomes industrial wastewater. The toxicities of vanadium compounds depend on the valence state and solubility of vanadium, among which V(V) is recognized as the most toxic and mobile species [5,6]. Vanadium is primarily present in the form of V(V) in oxidative aqueous media, which may cause hazardous effects on human health and the surrounding environment [7]. Therefore, it is essential to remove V(V) from wastewater to eliminate potential hazards.

Common technologies used to remove V(V) from aqueous solution include chemical precipitation [8], adsorption [9,10], ion exchange [11,12,13], solvent extraction [14,15], and membrane separation [16]. Conventional precipitation methods are ineffective for the treatment of the wastewater with a relatively low concentration of vanadium [17]. Solvent extraction and membrane separation may be accompanied by secondary pollution of organic solvents and high operating costs, respectively. Instead, adsorption and ion exchange have been widely used in water pollution control due to their high efficiencies and simple operations, which are suitable for the separation of heavy metal ions [18,19]. Various adsorbents have been employed to separate V(V) ions from solution, such as ion-exchange resins [20], nanocomposites [19], metal (hydr)oxides [21], and metal-organic frameworks (MOFs) [22]. Compared with other materials, ion-exchange resins exhibit high affinities and selectivities for targeted ions due to their abundant functional groups containing different donor atoms (such as N, O, S, and P) [23]. Moreover, low production costs and mature synthesis procedures allow resins to be utilized on a large scale.

To this end, it is necessary to find a suitable resin with special functional groups that can efficiently capture V(V) from the solution. Many studies on adsorbents bearing amine functional groups for the removal of V(V) have been reported. For example, Fan et al. found that the weak-base anion exchange resin (D314) with tertiary amines showed higher selectivity toward V(V) than Cr(VI), in both batch and column tests [24]. Anirudhan et al. studied the adsorption process of V(V) using a cellulose-based anion exchanger bearing secondary amines that exhibited a high adsorption capacity and excellent regeneration property [25]. An amine-functionalized polymer-grafted tamarind fruit shell was also evaluated by Anirudhan et al. for its efficiency in V(V) removal and recovery from simulated groundwater [26]. Several commercial ion exchange resins were tested to determine the optimum adsorption capacity for V(V) by Li et al., and the results indicated that the ZGA414 resin with tertiary amines exhibited better adsorption of V(V) than the strong-base ion exchange resin D202 [11]. However, traditional ion-exchange resins are usually accompanied by poor adsorption rate, high column pressures, and weak mechanical strengths because of their large particle sizes, fewer pores, and a high degree of swelling [27]. These factors are obstacles for rapid and efficient separation processes, making them unsuitable for practical applications.

In this work, a silica-supported primary amine resin (SiPAR) was synthesized by in situ polymerization in the pores of macroporous silica. The silica as the framework not only provided abundant pores for the composite resin but also restricted and protected the internal organic polymers, which resulted in fast adsorption kinetics, negligible swelling, and good mechanical properties [28]. To the best of our knowledge, there have been no studies on V(V) removal using this resin. The as-prepared material was characterized, and its adsorption performance for V(V) was investigated using batch experiments. In addition to the experimental analysis, a particle diffusion model was utilized to evaluate the real kinetics in the V(V) adsorption process.

## 2. Results and Discussion

### 2.1. Characterization

Figure 1a−d show the morphologies and elemental distributions of the SiPAR particles before and after V(V) adsorption. The samples exhibited uniform spherical morphologies, and their sizes were primarily in the range of 75−150 μm, which was the size range of the SiO_2_ particles. Furthermore, the EDS analysis indicated that the organic polymers were successfully loaded on the SiO_2_ based on the distribution of C. The presence of N element directly corresponded to the primary amine functional groups. The V(V) adsorbed by the SiPAR was also evident in the corresponding V elemental map.

The N_2_ adsorption-desorption isotherms and the pore diameter distribution of SiPAR are shown in Figure 1e. The isotherm curves were type-IV, suggesting that the resin was a mesoporous material. Moreover, the SiPAR had a narrow pore width distribution of approximately 60 nm. The structural properties of the three samples are shown in Table 1. The pore volume and average pore diameter of the SiPAR were significantly lower than those of the pure SiO_2_ particles, revealing that the polymers entered the SiO_2_ pores. Compared with the D302 resin, the SiPAR possessed a larger specific surface area, which provided more available active sites for the adsorbate and contributed to faster sorption.

Figure 1f shows the FT-IR spectra of the SiO_2_/SiPAR (before and after V(V) adsorption). The peaks at 3341 and 1640 cm^−1^ were related to the hydroxyl stretching and bending vibrations of adsorbed water, respectively. The bands at 1107, 798, and 497 cm^−1^ belonged to the characteristic peaks of SiO_2_ [29]. Moreover, the weak adsorption band at 2925 cm^−1^ in the SiPAR samples corresponded to the C−H of methylene groups [30], implying the presence of organic polymers. Furthermore, the peak at 2360 cm^-1^ might result from the adsorbed CO_2_ in the samples. There were no evident peaks of the primary amine in the spectrum, which might have been due to the overlap of NH_2_ and OH stretching vibrations [31]. In addition, the adsorption peak of V−O vibrations was found at 960 cm^−1^ [30]. These results indicated that V(V) ions were captured by the SiPAR from the solution.

The elemental analyzer was employed to evaluate the content of each element (C, H, and N) in the SiPAR. The mass percentage of N was 1.1%, which indicated that roughly 0.786 mmol/g of the primary amine groups were anchored onto the SiPAR. Furthermore, the mass percentages of C and H were 16.5% and 1.907%, respectively, which were relative to the content of the polymer loaded on the SiO_2_ particles.

Figure 1g,h show the TG-DSC curves of the SiPAR along with the silica-supported polystyrene (SiPS) for comparison. For the SiPAR, there was an endothermic effect in the range of 70–200 °C, corresponded to the mass loss of 0.46%, which resulted from the loss of adsorbed water; a weight loss about 1% at 200–300 °C was due to the presence of primary amine functional groups. The endothermic effect at 300–400 and 400–600 °C, separately corresponding to the thermooxidative decomposition of the benzene ring and carbon chain, were observed for both the SiPAR and SiPS [28]. These observations indicated that the functional groups were successfully grafted to the silica-based polymers.

In addition, acid-base titration was employed to determine the total exchange capacity of the SiPAR, and the obtained results suggested 0.66 meq/g of primary amine functional groups were available. The related steps of titration are described in Appendix A.

### 2.2. Batch Studies

#### 2.2.1. Effect of Initial pH

Solution pH could impact the V(V) form in the aqueous phase and the surface charge of adsorbent, which might further influence the adsorption process of V(V). As shown in Figure 2b, the removal efficiency of V(V) rose as the pH increased from 2 to 4, and it declined gradually when the pH was above 4. The optimum adsorption pH was 4, and the corresponding maximum removal efficiency was 99.33%. Apparently, both the adsorbent and adsorbate were closely affected by solution pH. First, the adsorbent with primary amine groups must be protonated by combining with H^+^ before adsorbing V(V). The protonation process was gradually subdued as the pH value increased, and pentavalent vanadium had a very complicated species distribution in the aqueous solution under different pH conditions. Figure 2a shows the speciation and proportion of V(V) at various pH values. The cationic VO_2_^+^ was the most prevalent form of V at pH = 2, and the second was anionic H_3_V_2_O_7_^−^, coexisting with a small amount of H_3_VO_4_ and negligible amounts of other species. As the pH value increased to 4, V(V) was almost entirely present as H_3_V_2_O_7_^−^. Subsequently, the proportion of H_3_V_2_O_7_^−^ decreased, and the content of other anions (e.g., H_2_VO_4_^−^, V_3_O_9_^3−^, V_4_O_12_^4−^, HVO_4_^2−^, and HV_2_O_7_^3−^) increased at pH values higher than 4. Based on the above analyses, it was concluded that poor interaction between the protonated amino groups and VO_2_^+^ weakened the adsorption efficiency when the pH was lower than 4. However, H_3_V_2_O_7_^-^, as the predominant specie of V(V) at pH = 4, allowed to remove more vanadium per active site than other forms of the V(V) ions. These can be described as usind Equations (1)–(4):(1)$${\mathrm{RNH}}_{3}^{+}{\mathrm{Cl}}^{-}{+H}_{3}V_{2}O_{7}^{-}\to {\mathrm{RNH}}_{3}^{+}H_{3}V_{2}O_{7}^{-}+{\mathrm{Cl}}^{-}$$
(2)$${\mathrm{RNH}}_{3}^{+}{\mathrm{Cl}}^{-}{+H}_{2}{\mathrm{VO}}_{4}^{-}\to {\mathrm{RNH}}_{3}^{+}H_{2}{\mathrm{VO}}_{4}^{-}+2{\mathrm{Cl}}^{-}$$
(3)$${2\mathrm{RNH}}_{3}^{+}{\mathrm{Cl}}^{-}{+\mathrm{HVO}}_{4}^{2-}\to {\left({\mathrm{RNH}}_{3}^{+}\right)}_{2}{\mathrm{HVO}}_{4}^{2-}+{\mathrm{Cl}}^{-}$$
(4)$${3\mathrm{RNH}}_{3}^{+}{\mathrm{Cl}}^{-}{+\mathrm{HV}}_{2}O_{7}^{3-}\to {\left({\mathrm{RNH}}_{3}^{+}\right)}_{3}{\mathrm{HV}}_{2}O_{7}^{3-}+3{\mathrm{Cl}}^{-}$$

Therefore, both the suppression of the protonation process and the decrease in the H_3_V_2_O_7_^-^ proportion at higher pH values led to a decrease in V(V) removal efficiency as the pH value increased.

#### 2.2.2. Sorption Kinetics

In order to investigate the effect of contact time, the kinetics study of V(V) on the SiPAR and D302 resin was conducted. Figure 2c shows the V(V) adsorbed amounts of two resins at the various contact times. The uptake amounts of V(V) raised as the contact time increased, and a plateau was reached within 90 min for the SiPAR, while it was more than 180 min for the D302. The SiPAR showed fast adsorption kinetics compared with the commercial resin, which was ascribed to the bigger specific surface area and lesser particle size, enhancing the mass transfer process. 

In addition, the kinetics data were interpreted by the pseudo-first-order kinetic model (Equation (5)) and the pseudo-second-order kinetic model (Equation (6)):

Pseudo-first-order:(5)$$Q_{t}=Q_{e}\left(1-e^{-k_{1}t}\right)$$

Pseudo-second-order:(6)$$t/Q_{t}=1/k_{2}Q_{e}^{2}+t/Q_{e}$$
where *Q_e_* and *Q_t_* (mg/g) separately represent the uptake of V(V) at adsorption equilibrium and time t (min), *k*_1_ (min^−1^) and *k*_2_ (g mg^-1^ min^−1^) denote the rate constant of pseudo-first-order and pseudo-second-order kinetics. The fitted kinetics parameters are given in Table 2. For the pseudo-second-order kinetics model, its values of correlation coefficients (R^2^) were comparatively higher than those of the pseudo-first-order kinetics model, which could be used to satisfactorily describe the kinetic data of both resins. This demonstrated that chemical adsorption was dominant in the sorption process of two resins for V(V).

#### 2.2.3. Sorption Isotherm

Figure 2d shows the V(V) adsorption isotherm onto the SiPAR at 298 K. The sorption amounts of V(V) enhanced as the initial concentration increased, and the maximum uptake amount of V(V) was 70.57 mg/g when the initial V(V) concentration was 400 mg/L. To investigate the behaviors that resins interacted with solutes in the sorption process, isotherm models were applied to simulate the V(V) sorption curve, for which Langmuir, Freundlich, and Redlich–Peterson models were widely applied. Their mathematic representations are expressed in the Equations (7)–(9) as follows [26]:

Langmuir:(7)$$Q_{e}=Q_{m}K_{L}C_{e}/\left(K_{L}+C_{e}\right)$$

Freundlich:(8)$$Q_{e}=K_{F}C_{e}^{1/n}$$

Redlich–Peterson:(9)$$Q_{e}=K_{R}C_{e}/\left(1+a_{R}C_{e}{}^{b_{R}}\right)$$
where *Q_e_* (mg/g) and *C_e_* (mg/L) separately denote the equilibrium uptake of V(V) and the corresponding equilibrium concentration; *Q_m_* (mg/g) represents the saturated sorption amounts; *K_L_* (L/mg) denotes the Langmuir adsorption constant; *K_F_* and n represent the constants related to Freundlich isotherm; *K_R_*, *a_R_* , and *b_R_* denote the Redlich–Peterson constants. The relevant parameters are listed in Table 3. The result presented that Redlich–Peterson model provided the best fit to the experimental data due to the highest R^2^ (0.995) among the three models. Moreover, the *Q_e_* value calculated using the model nearly agreed with the experimental value. Therefore, the sorption process was predominantly heterogeneous multilayer adsorption, as the fitting curve was closer to the Freundlich isotherm.

A comparison of the V(V) adsorption performance of SiPAR with other adsorbents reported in previous studies is shown in Table 4. The SiPAR exhibited moderate uptake amounts and had favorable sorption kinetics when compared with most of the previously reported materials. 

#### 2.2.4. Effect of Coexisting Ions

A variety of anions are typically present in wastewater, which may compete with V(V) for the active sites on the adsorbents, impacting the removal efficiency for the targeted ions. Consequently, it is necessary to investigate the selective removal of the SiPAR for V(V) anions coexisting with other anions. Several common anions—Cl^−^, SO_4_^2−^, and NO_3_^−^—were introduced as interfering ions to assess the adsorption properties of the SiPAR. Figure 2e presents the influence of coexisting anions on the V(V) removal efficiency of SiPAR. The three kinds of anions with high concentrations had a negligible influence on the V(V) uptake. As the salt concentration increased ranging from 5 to 100 mmol/L, the removal efficiency only decreased from 99.3% to 97.5% (for SO_4_^2−^). More than 97% of the adsorption occurred with a salt concentration of 100 mmol/L, which exceeded the molar quantity of V(V) by 50-fold. Therefore, it was concluded that the SiPAR exhibited excellent resistance to common anions, which implied that this material is promising for application in wastewater for the selective capture of V(V).

#### 2.2.5. Sorption Thermodynamics

To explore the influence of temperature on the V(V) adsorption, the adsorption thermodynamics of the SiPAR for V(V) was investigated at various temperatures (298, 308, 318, and 328 K). The Gibbs free energy change (Δ*G*), enthalpy change (Δ*H*), and entropy change (Δ*S*) were calculated as follows:(10)$$\Delta G=-RT\ln K_{d}$$
(11)ΔG=ΔH−TΔS
where *K_d_* (L/mg) represents the distribution coefficient related to the temperature *T* (K), and R (8.314 J mol^−1^ K^−1^) denotes the universal gas constant. The values of ln*K_d_* versus 1/T are plotted and presented in Appendix A, and the thermodynamic parameters are shown in Appendix A. The Δ*G* was a negative value, indicating that the V(V) sorption by the SiPAR was a spontaneous process. The value Δ*H* was 8.96 kJ/mol, which suggested the process was endothermic, and the higher temperature was favorable for adsorption. The value Δ*S* was 73.38 J/(mol K), indicating that the randomness enhanced at the solid-solution interface.

### 2.3. Mathematical Modeling

In batch experiments, the pseudo-first-order and pseudo-second-order kinetics models are commonly used to describe the sorption process, but they cannot provide any information about the real sorption rate. Based on the theory described in Section 3.5, the *k_f_* and *D_p_* were determined by fitting the adsorption curves of V(V) with particle diffusion models. Accordingly, the real kinetics rate based on the mass transfer resistance could be predicted. The experimental decay concentration curves of V(V) on the two resins and the corresponding model fits for the ion exchange kinetics are illustrated in Figure 3a,b. The SiPAR exhibited higher values of *k_f_* (1.1 × 10^−5^ m/s) and D_p_ (6.7 × 10^−9^ m^2^/s) than those of the D302 resin (*k_f_* = 6.58 × 10^−6^ m/s, D_p_ = 4.7 × 10^−9^ m^2^/s), which corresponded to a faster sorption rate, in agreement with the results above.

On the basis of the values of *k_f_* and *D_p_*, it was concluded that film mass transfer was significantly faster than the intraparticle diffusion, meaning that the sorption process of V(V) onto the resin was primarily controlled by intraparticle diffusion. Consequently, to further illustrate the diffusion of V(V) inside the resin, the dimensionless V(V) concentration change in the radial direction inside two resin particles at various time periods was predicted by the intraparticle diffusion model. Figure 3c,d show that the time required for V(V) diffusion from the resin surface to the center was approximately 15 min for the SiPAR but more than 60 min for the D302 particle at the same conditions. In addition, approximately 90 min was required for the V(V) concentration to approach saturation at the center of SiPAR resin. However, for the D302 resin, the time was more than 180 min. These simulated results confirmed that the SiPAR had a faster mass-transfer rate.

## 3. Material and Methods 

### 3.1. Materials

Macroporous silica (SiO_2_) with a pore size of 75–150 μm and a 69% porosity was utilized as the base material to load the organic polymers. Monomers for the polymerization procedure were purchased from Shanghai Aladdin Bio-Chem Technology Co., Ltd. (Shanghai, China). The styrene (St) was of 99% purity and contained a polymerization inhibitor; the divinylbenzene (DVB) was an m/p- mixture with 55% purity, both of which required further purification before the experiments of resin synthesis. Sodium orthovanadate (Na_3_VO_4_, Shanghai Macklin Biochemical Co., Ltd, Shanghai, China) was used as the source of fV(V) in the experiments. The other chemicals were obtained from the Guangzhou Jinhuada Chemical Reagent Co., Ltd, Guangzhou, China. All of the reagents were of analytical grade unless stated otherwise. Ultrapure water (18.2 MΩ, Hitech, Shanghai, China) was employed to prepare a solution. D302 resin was purchased from Zhengzhou Hecheng New Material Technology co. LTD (Henan province, China) and acted as a comparison material for the silica-supported resin in the kinetics experiment. In addition, D302 resin was pretreated according to the descriptions provided by the manufacturers before adsorption. Their relevant information is summarized in Table 5.

### 3.2. Synthesis of SiPAR

The main synthesis steps of the SiPAR are given in Figure 4 and are described as in the following subsections:

#### 3.2.1. Preparation of Silica-Supported Polystyrene (SiPS)

SiPS was synthesized by an in situ polymerization method, where styrene and divinylbenzene were drawn into the silica pores in a rotating flask under a vacuum state and then reacted under suitable conditions. The relevant details are described in a previous purification [28].

#### 3.2.2. Preparation of Silica-Supported Chloromethylated Polystyrene (SiPS-CH_2_Cl)

First, 40 g of SiPS was immersed in 120 mL of chloromethyl methyl ether and stirred at room temperature for 3 h. Next, 7.2 g of anhydrous zinc chloride particles was added as a catalyst in two batches and started reacting under agitation at 313 K. After 12 h, the products were filtered from the mixtures, then washed with alcohol and deionized water alternatively, and, finally, dried in a vacuum oven. 

#### 3.2.3. Preparation of Silica-Supported Primary Amine Resin (SiPAR)

A mixture of 10 g of SiPS-CH_2_Cl and 40 mL of trichloromethane was stirred at room temperature for 3 h and then reacted with excess hexamethylenetetramine at 308 K for 11 h under constant agitation. The resulting hexamethylenetetramine-grafted SiPS-CH_2_Cl was washed several times with alcohol and then added to a mixture of hydrochloric acid and ethanol (volume ratio 1:3) at 313 K for 2 h while stirring. During this process, the Delépine reaction occurred, and the hexamethylenetetramine anchored in the SiPS-CH_2_Cl was decomposed to generate primary amines (SiPAR). The as-prepared materials were repeatedly washed using alcohol and deionized water and, finally, thoroughly dried in a vacuum oven.

### 3.3. Characterization

The surface structure and elemental analyses of the materials were conducted by field-emission scanning electron microscopy with energy-dispersive X-ray spectroscopy (FESEM−EDS, Hitachi, Japan). The Brunauer–Emmett–Teller surface areas and pore structures were tested using a specific surface area and porosity analyzer (BET, Tristar II 3020, Norcross, GA, USA). The functional groups in the samples were detected by Fourier-transform infrared spectroscopy (FT-IR, Shimadzu IR Tracer 100, Japan). Thermal analyses of the samples were conducted by thermogravimetry and differential scanning calorimetry (TG-DSC, Netzsch STA449F3, Selb, Germany). The elemental compositions of the samples were determined by the elemental analyzer (EA, Elementar Analysensystem, Langenselbold, Germany). 

### 3.4. Batch Experiments

Batch mode tests were performed to investigate the adsorption performance of the SiPAR for V(V) and determine the optimal experimental conditions. In this procedure, 0.05 g of resins were added to 20 mL of solution in 45-mL glass bottles, which were placed in a water bath shaker (298 K, 120 r/min) to perform the experiments. The pH values of all solutions were set to 4, except when the influence of the solution pH was explored. The pH of all solutions was adjusted by adding 1 M HCl or 1 M NaOH. Each batch test was performed for 3 h except when the influence of the contact time was studied. Furthermore, the initial concentration of the solutions was set to 100 mg/L for all experiments except when the impact of the concentration on the uptake amount was researched. The influence of the solution pH was investigated by varying the initial pH value from 2 to 11. To compare the sorption kinetics of the two resins, the removal efficiencies of V(V) ions by the SiPAR and D302 were determined at time intervals of 10, 20, 40, 60, 90, 120, and 180 min. To obtain the isotherms, the experiments were carried out by changing V(V)’s initial concentrations that range from 50 to 400 mg/L. Thermodynamic tests were conducted by changing the experimental temperatures (298, 308, 318, and 328 K). After the termination of each reaction, the mixture was separated through a 0.45 μm membrane filter, and the obtained solution was diluted based on the experimental conditions. Finally, the solutions were analyzed using inductively coupled plasma−atomic emission spectroscopy (ICP−AES, Shimadzu ICPS-7510, Kyoto, Japan) to determine the V(V) concentration in the solution. The adsorption capacity *q_e_* (mg/g) was calculated using Equation (12):(12)$$q_{e}=\left(C_{0}-C_{e}\right)\times V/m$$
where *C*_0_ and *C_e_* (mg/L) are the concentrations of V(V) before adsorption and after equilibration, respectively, *V* (mL) is the volume of the solution, and *m* (g) is the mass of resin used. The removal efficiency E% was calculated with Equation (13):(13)$$E\%=\left(C_{0}-C_{t}\right)/C_{0}\times 100\%$$
where C_t_ is the V(V) concentration at time t (min). In order to decrease the experimental error and ensure the accuracy of data, all tests were conducted in duplicate.

### 3.5. Mathematical Models

The process that V(V) ions were adsorbed on resin particles from aqueous solution involved three procedures: external film mass transfer, intraparticle diffusion, and ion exchange. For an ion-exchange process, the adsorbate was first transferred from the solution to the external liquid film of the resin granules due to a concentration gradient. The adsorbate then needed to diffuse inside the resin from the liquid film. Finally, the ion exchange between the active sites in the particle pores and adsorbate ions achieved successful adsorption [37]. It is generally assumed that the ion-exchange reactions between the adsorbed ions and functional groups are instantaneous, and thus, the sorption process is restricted by film transfer and intraparticle diffusion. Consequently, the particle diffusion model was adopted to investigate the real sorption rate [38], where the relevant parameters, the film mass-transfer coefficient *k_f_* (m/s), and pore diffusivity *D_p_* (m/s^2^) were also calculated.

Based on mass balance for the single resin, the partial differential equations are represented as follows [37]:(14)$$V\frac{dC}{dt}+\frac{3m}{\rho_{p}R_{p}}k_{f}\left(C-C_{p}\left|{}_{R=R_{p}}\right.\right)=0$$
(15)$$\varepsilon_{p}\frac{\partial C}{\partial t}+\rho_{p}\frac{\partial q}{\partial t}=\frac{1}{R^{2}}\frac{\partial }{\partial R}\left(D_{p}R^{2}\frac{\partial C}{\partial R}\right)$$
where *C* and *C_p_* (mol/m^3^) represent the V(V) concentration in the bulk solution and inside the resin phase, respectively, *V* (m^3^) denotes the solution volume, *t* (min) denotes the time, ε_p_ is the particle porosity, *m* and *ρ_p_* are separately the weight (g) and an apparent density (g/m^3^) of the resin granules, *R_p_* (m) is the resin particle radius, *q* (mol/g) is the V(V) adsorbed amount by the resin, R is the distance along the radius direction.

The initial condition is:(16)$$C(R,0)=C_{0},C_{p}\left(R,0\right)=0$$
and boundary conditions are:(17)$$\frac{\partial C_{p}}{\partial R}\left|{}_{R=0}\right.=0$$
(18)$$\frac{\partial C_{p}}{\partial R}\left|{}_{R=R_{p}}\right.=\frac{k_{f}}{D_{p}}\left(C-C_{p}\left|{}_{R=R_{p}}\right.\right)$$

To solve Equations (14)–(18) simultaneously with the corresponding initial and boundary conditions, the finite element method was employed using COMSOL Multiphysics (4.4, Shenzhen University, Shenzhen, China).

## 4. Conclusions

In this work, a silica-supported primary resin was prepared to remove V(V) from solutions. The physicochemical characteristics of the resin were thoroughly explored by SEM-EDS, BET, FT-IR, and EA. The kinetics study indicated that the silica-based resin exhibited a faster adsorption rate than the commercial resin (D302), which was ascribed to its bigger specific surface area and smaller particle size. Moreover, mathematical models were applied to further investigate adsorption progress. The V(V) film mass transfer coefficient (*k_f_*) and pore diffusivity (*D_p_*) onto the SiPAR were estimated to be 1.1 × 10^−5^ m/s and 6.7 × 10^−9^ m^2^/s, respectively, and the intraparticle diffusion was the rate-controlling step. Compared with the D302 resin, SiPAR showed superiority in the intraparticle diffusion process, contributing to the prominent kinetics rate. In conclusion, the prepared SiPAR is expected to be able to rapidly and selectively capture V(V) from wastewater on a large scale.

## Figures and Tables

**Figure 1 molecules-25-01448-f001:**
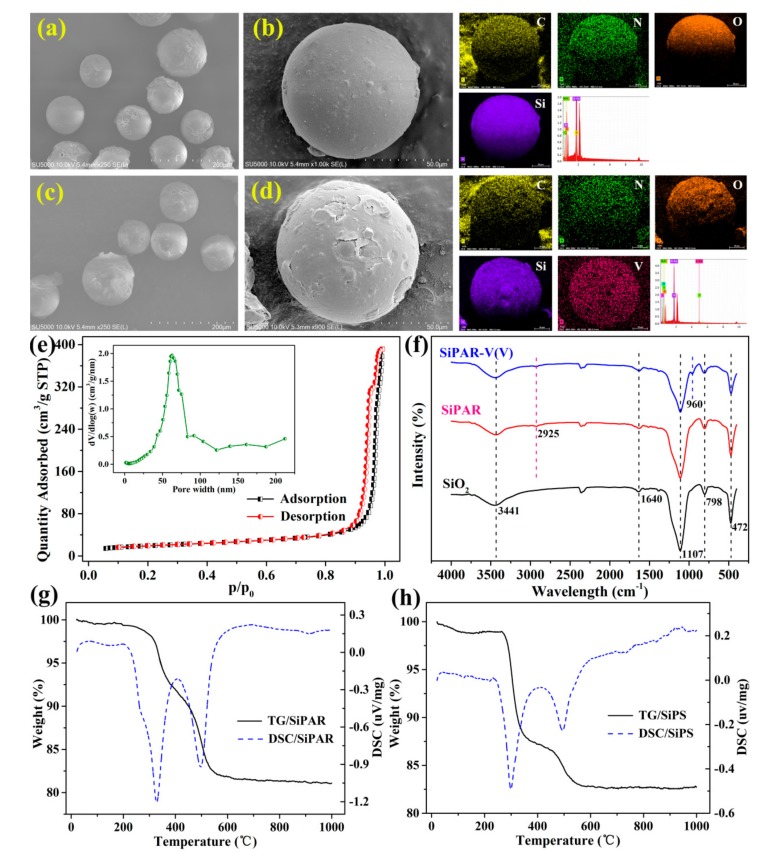
Relevant characterization: (**a**) SEM micrograph and EDX spectra for the prepared silica-supported primary amine resin (SiPAR) with a 250× magnification and (**b**) the corresponding EDS results with a 1000× magnification, (**c**) the SiPAR after V(V) adsorption with a 250× magnification and (**d**) the corresponding EDS results with a 1000× magnification, (**e**) the N_2_ adsorption-desorption isotherms and pore width distribution of the SiPAR, (**f**) FT-IR spectra of SiO_2_, SiPAR, and SiPAR-V(V) (**PS**: SiPAR-V(V) denotes the SiPAR after V(V) adsorption.), (**g**) TG-DSC curves of the SiPAR and (**h**) the SiPS.

**Figure 2 molecules-25-01448-f002:**
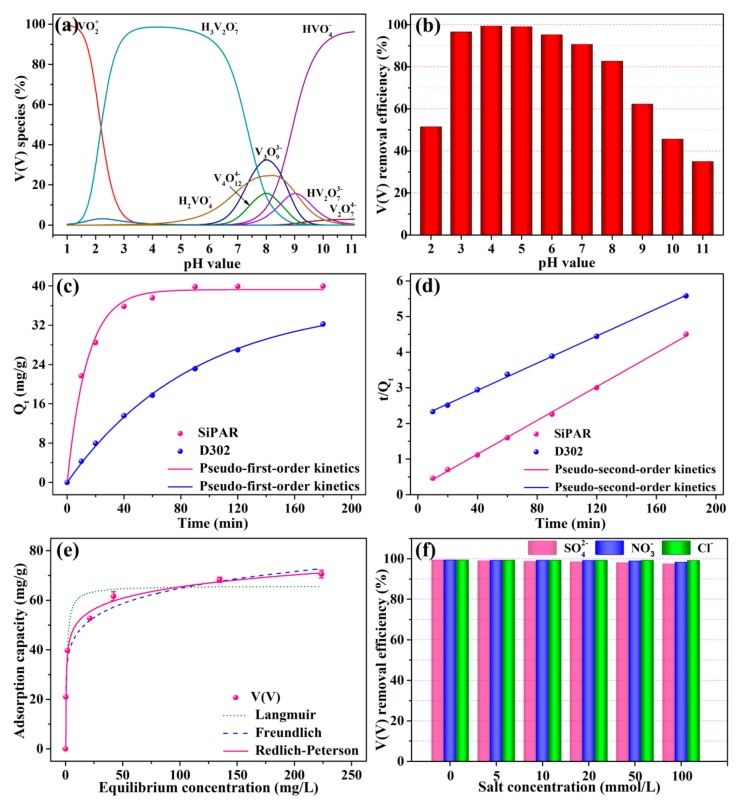
Batch adsorption experiments: (**a**) V(V) ionic species distribution at various pH with 100 mg/L total V(V) concentration, (**b**) removal efficiency of V(V) in the pH range from 2 to 11, (**c**) Pseudo-first-order kinetics and (**d**) pseudo-second-order kinetics fits for the V(V) kinetics data of the SiPAR and D302 resins, (**e**) V(V) adsorption isotherm onto the SiPAR, (**f**) effect of coexisting anions on V(V) removal efficiency of SiPAR (aqueous volume: 20 mL, resin mass: 0.05 g, T = 298 K). **PS**: V(V) form was calculated by PHREQQ, which is a software used to evaluate the speciation and proportion of complex chemicals in the solution.

**Figure 3 molecules-25-01448-f003:**
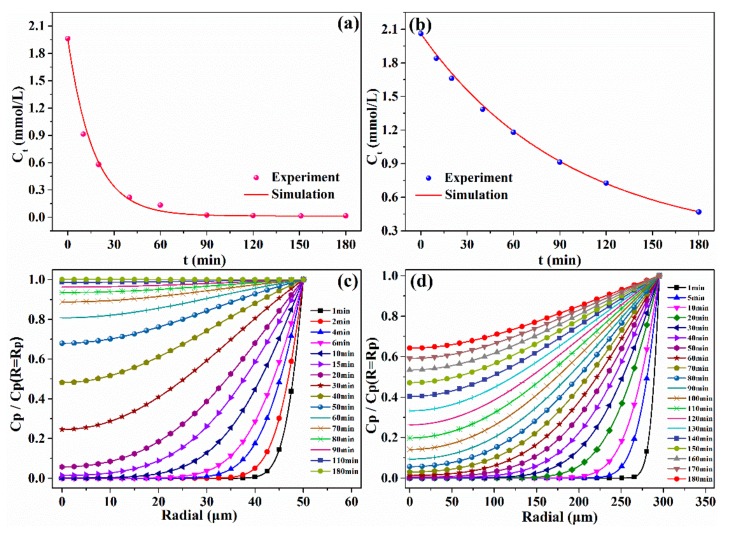
Mathematical modeling of kinetics data: Uptake curves of the (**a**) SiPAR and (**b**) D302 resins fitted by the particle diffusion model, and dimensionless V(V) concentration distributions in the radial direction inside the resin at different times for the (**c**) SiPAR and (**d**) D302 resin.

**Figure 4 molecules-25-01448-f004:**
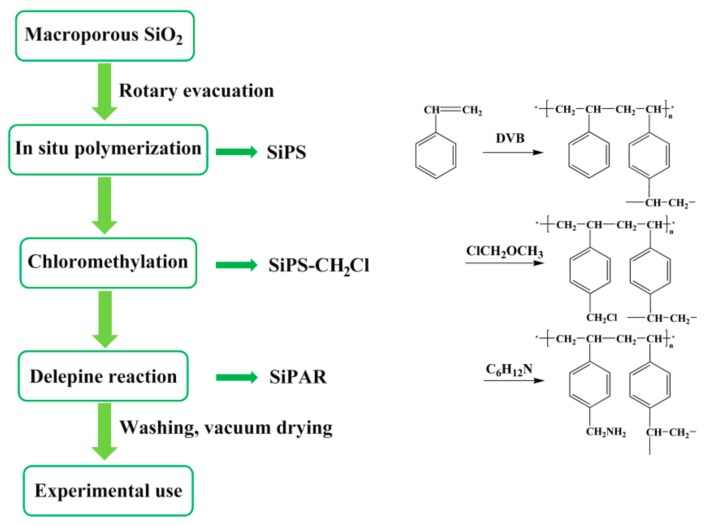
Procedure for the synthesis of SiPAR.

**Table 1 molecules-25-01448-t001:** Comparison of structural properties for three samples.

Sample	Surface Area (m^2^/g)	Average Pore Diameter (nm)	Pore Volume (mL/g)
SiO_2_	80.4	50.3	1.02
SiPAR	68.31	41.22	0.60
D302	30.36	33.43	0.25

**Table 2 molecules-25-01448-t002:** Parameters calculated from the pseudo-first-order and pseudo-second-order kinetics.

Resin	Pseudo-First-Order			Pseudo-Second-Order		
	*Q*_e_(mg/g)	*k*_1_(min^−1^)	*R* ^2^	*Q*_e_ (mg/g)	*k*_2_(g/mg min)	*R* ^2^
SiPAR	39.26	0.0704	0.9926	40.25	0.02	0.9990
D302	36.84	0.0112	0.9986	52.49	0.00017	0.9989

**Notes**: *Q*_e_ represents the uptake of V(V) at adsorption equilibrium, *k*_1_ and *k*_2_ denote the rate constant of pseudo-first-order and pseudo-second-order kinetics, *R*^2^ is the correlation coefficients.

**Table 3 molecules-25-01448-t003:** Parameters of adsorption isotherm calculated from different models.

Langmuir			Redlich–Peterson				Freundlich		
*Q* _m_	*K_L_*	*R* ^2^	*K* _R_	*a* _R_	*n*	*R* ^2^	*K* _F_	*n*	*R* ^2^
mg/g	L/mg		mg/g				L^n^/mg^(n−1)^		
64.17	0.836	0.93	214.17	5.19	0.899	0.995	33.51	6.98	0.979

**Notes**: *Q*_m_ (mg/g) represents the saturated sorption amounts; *K*_L_ denotes the Langmuir adsorption constant; *K*_F_ and n represent the constants related to Freundlich isotherm; *K*_R_, *a*_R_, and *b*_R_ denote the Redlich–Peterson constants, *R*^2^ is the correlation coefficients.

**Table 4 molecules-25-01448-t004:** Comparison of sorption properties with other adsorbents for V(V).

Adsorbent	C_0_ (mg/L)	Adsorption Capacity	Equilibrium Time	Reference
MZ-PPY	25–250	65 mg/g	About 90 min ^a^	[7]
Zr(IV)-SOW	-	51.1 mg/g	20 h	[18]
Cell-AE	25–600	197.75 mg/g	1 h	[25]
PGTFS–NH_3_^+^Cl^−^	10–300	45.86 mg/g	4 h	[26]
ZrO_2_	-	54.3 mg/g	24 h	[32]
Waste metal sludge	7.6–48.4	24.8 mg/g	7 h	[33]
CCSB	20–150	148.15 mg/g	9 h	[34]
GTMAC	10-591	34.3 mg/g	24 h	[35]
TiO_2_ nanoparticles	3-800	50 mg/g	30 min	[36]
D302	700	4 mg/g	>3 h	This work
SiPAR	50–400	70.57 mg/g	90 min	This work

Notes: “-” means that the property was not provided; “a” denotes that the value was observed from the kinetics figure.

**Table 5 molecules-25-01448-t005:** Detailed information of two resins.

Resins	SiPAR	D302
Matrix	St-DVB	St-DVB
Physical form	Spherical bead	Spherical bead
Average particle diameter	100 µm	590 µm
Functional groups	R–NH_2_	R–NH_2_
Resin particle porosity, ε_p_	72.7%	8.6%
Resin particle apparent density, ρ_p_	1.6426 × 10^3^ kg/m^3^	1.0421 × 10^3^ kg/m^3^
Total exchange capacity	0.66 meq/g	3 meq/g

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
