# Peer review of "Removal of V(V) From Solution Using a Silica-Supported Primary Amine Resin: Batch Studies, Experimental Analysis, and Mathematical Modeling"

_molecules, 2020, doi:10.3390/molecules25061448_

Round 1
Reviewer 1 Report
The manuscript reports in situ synthesis of an anion-exchange resin within pores of macroporous silica beads and its use for decontamination of water polluted by V(V) species. The topic is interesting and the adsorption part is described fairly well with convincing description of kinetics and of adsorption isotherm. What is essentially missing is the characterization of the amount and structure of resin inside the beads. FTIR is not convincing at all, showing practically no signals from the polymer. The spectra contain a visible band of CO2, which means that background correction was not made properly. For quantitative characterization of the polymer it would be helpful to get TGA for the material and also complement this by pH-titration to determine the content of amino groups.
The thermodynamic data for adsorption are very peculiar: endothermic effect of complexation and positive enthropy effect of adsorption are unusual. The authors need at least to present a plausible hypothesis for this. Otherwise, the electrostatic interaction of poorly solvated vanadate species should be expected to provide negative enthalpy change.
Author Response
Reviewer #1: The manuscript reports in situ synthesis of an anion-exchange resin within pores of macro porous silica beads and its use for decontamination of water polluted by V(V) species. The topic is interesting and the adsorption part is described fairly well with convincing description of kinetics and of adsorption isotherm. What is essentially missing is the characterization of the amount and structure of resin inside the beads. FT-IR is not convincing at all, showing practically no signals from the polymer. The spectra contain a visible band of CO2, which means that background correction was not made properly. For quantitative characterization of the polymer it would be helpful to get TGA for the material and also complement this by pH-titration to determine the content of amino groups.
Author’s reply: We first wish to thank Reviewer 1 a lot for having undertaken review of our manuscript and giving the constructive comments. We have added the relevant characterization of TG-DSC for the SiPAR in Fig. 2d, and it is hard to observe the structure of resin inside the macroporous SiO2 by SEM for the poor pores size; if you have an interest in the section of the silica-supported resin, the reference [Wang, X.; Ye, Z.; Chen, L.; Zheng, Q.; Liu, C.; Ning, S.; Khayambashi, A.; Wei, Y. Microporous silica-supported cation exchanger with superior dimensional stability and outstanding exchange kinetics, and its application in element removal and enrichment. React. Funct. Polym. 2019, 142, 87–95.] can provide the relevant information.
[Page 7, line 248−256]: “……Fig. 2g and h show the TG-DSC curves of the SiPAR along with the SiPS for comparison……”
We have corrected the background carefully, and the occurrence of CO2 band was resulted from the mold. In fact, it doesn’t matter with the results. We accepted the reviewer’s good suggestion, and had clarified the signals from the polymer in the paper (“Moreover, the weak adsorption band at 2925 cm−1 in the SiPAR samples corresponded to the C−H of methylene groups [32], implying the presence of organic polymers. There were no evident peaks of the primary amine in the spectrum, which may have been due to the overlap of NH2 and OH stretching vibrations [33]. In addition, the adsorption peak of V−O vibrations was found at 960 cm−1 [32]. These results indicate that V(V) ions were captured by the SiPAR from solution.”). On the basis of the above description, FT-IR analysis confirmed the presence of V(V) within the structures of the SiPAR after the sorption process.
Following the reviewer’s kind advices, we have added the section of pH-titration to determine the content of amino groups, which we hope will meet with his/her approval.
[Page 7, line 257−260]: “In addition, acid-base titration was employed to determine the total exchange capacity of the SiPAR, and the obtained results suggested 0.66 meq/g of primary amine functional groups were available. The related steps of titration were described in Supplementary data.”
The thermodynamic data for adsorption are very peculiar: endothermic effect of complexation and positive enthropy effect of adsorption are unusual. The authors need at least to present a plausible hypothesis for this. Otherwise, the electrostatic interaction of poorly solvated vanadate species should be expected to provide negative enthalpy change.
Author’s reply: Thank Reviewer so much for the comments. We regret for not being able to explain the thermodynamic process, but we find the similar endothermic phenomena in other V(V) adsorption processes in some literatures. For example, Yeom et al. studied the effect of solution temperature on the adsorption V(V) onto the ion exchanger with amine functional group, and found the equilibrium adsorption capacity is slightly increased with the rising temperature. They thought solution temperature affected the electroselectivity reaction, ion salvation formation of ion pairs, and association and formation of complexes between vanadium (V) ions and the ion exchanger, which led to the results. Mthombeni et al. observed that the sorption of vanadium (V(V)) on MZ-PPY (Magnetized natural zeolite-polypyrrole) was favourable at high temperature, with the positive DH of 19.468 kJ/mol. Zhang et al. also found that the adsorption of V(V) ions onto chitosan-Zr(IV) composite was more favorable at high temperatures. [Yeom, B.-Y.; Lee, C.-S.; Hwang, T.-S. A new hybrid ion exchanger: effect of system parameters on the adsorption of vanadium (V). J. Hazard. Mater. 2009, 166, 415–420. Mthombeni, N.H.; Mbakop, S.; Ochieng, A.; Onyango, M.S. Vanadium (V) adsorption isotherms and kinetics using polypyrrole coated magnetized natural zeolite. J. Taiwan Inst. Chem. Eng. 2016, 66, 172–180.; Zhang, L.; Liu, X.; Xia, W.; Zhang, W. Preparation and characterization of chitosan-zirconium(IV) composite for adsorption of vanadium(V). Int. J. Biol. Macromol. 2013. (Artcle in press)]. Therefore, the thermodynamic data in this study are rational.
Reviewer 2 Report
The paper is well written with enough data as a scientific paper. I suggest publication after minor revision.
in the title, batch studies should be added.
The methodology needs to be cited by relevant papers.
Section 3.7 has a different format.
The ability of adsorption of V by these 2 types of resins should be compared with other types of resins in the previous studies in table form.
How much is the predicted cost of using these resins in large scale applications in comparison to other types of adsorbents? is there any estimation by others?
Author Response
Reviewer #2:
The paper is well written with enough data as a scientific paper. I suggest publication after minor revision.
in the title, batch studies should be added.
Author’s reply: Thank Reviewer very much for the valuable comments and suggestions. On the basis of reviewer’s good suggestions, we have revised the title of the manuscript as “Removal of V(V) from solution using a silica-supported primary amine resin: batch studies, experimental analysis and mathematical modeling”.
The methodology needs to be cited by relevant papers.
Author’s reply: Thank Reviewer a lot for these valuable comments and the detailed suggestions. We had made the necessary corrections accordingly by citing the related reference in the revised manuscript.
[Page 6, line 197]: “……Based on mass balance for the single resin, the partial differential equations are represented as follows [29]:……”
Section 3.7 has a different format.
Author’s reply: Thank Reviewer very much for the valuable comments. We had made the necessary corrections by revising the related subheadings in the section 3 according to reviewer’s kind advice, which we hope will meet with the Reviewer’s approval.
[Page 8, line 263]: “……3.2. Batch studies……”
The ability of adsorption of V by these 2 types of resins should be compared with other types of resins in the previous studies in table form.
Author’s reply: Thank Reviewer very much for the valuable comments. Following the suggestion of the reviewer, the adsorption amount of D302 resin has been added in the Table 5.
[Page 17, line 564]: “……Table 5……”
Table 5
Comparison of sorption properties with other adsorbents for V(V).
|
Adsorbent |
C0 (mg/L) |
Adsorption capacity |
Equilibrium time |
Reference |
|
MZ-PPY |
25–250 |
65 mg/g |
About 90 mina |
[7] |
|
Zr(IV)-SOW |
- |
51.1 mg/g |
20 h |
[18] |
|
Cell-AE |
25–600 |
197.75 mg/g |
1 h |
[25] |
|
PGTFS–NH3+Cl− |
10–300 |
45.86 mg/g |
4 h |
[26] |
|
ZrO2 |
- |
54.3 mg/g |
24 h |
[34] |
|
Waste metal sludge |
7.6–48.4 |
24.8 mg/g |
7 h |
[35] |
|
CCSB |
20–150 |
148.15 mg/g |
9 h |
[36] |
|
GTMAC |
10-591 |
34.3 mg/g |
24 h |
[37] |
|
TiO2 nanoparticles |
3-800 |
50 mg/g |
30 min |
[38] |
|
D302 |
700 |
302.4 mg/g |
>3 h |
This work |
|
SiPAR |
50–400 |
70.57 mg/g |
90 min |
This work |
Notes: “-” means that the property was not provided; “a” denotes that the value was observed from the kinetics figure.
How much is the predicted cost of using these resins in large scale applications in comparison to other types of adsorbents? is there any estimation by others?
Author’s reply: Thank Reviewer very much for the valuable comments. We has conducted a simple economic analysis about a silica-based pyridine resin (SiPyR-N4) to remove Cr(VI) in previous literature [Ye, Z.; Yin, X.; Chen, L.; He, X.; Lin, Z.; Liu, C.; Ning, S.; Wang, X.; Wei, Y. An integrated process for removal and recovery of Cr(VI) from electroplating wastewater by ion exchange and reduction–precipitation based on a silica-supported pyridine resin. J. Cleaner Prod. 2019, 236, 117631.]. The cost of SiPyR-N4 is approximately 43.3 USD/kg for industrial application. But for the SiPAR, there is no relevant prediction about the cost. Compared with the SiPyR-N4, the synthesis of SiPAR is relatively facile. So we roughly estimate the cost of the SiPAR was less than 43.3 USD/kg.
Reviewer 3 Report
This study described a Removal of V(V) from solution using a silica-supported primary amine resin. The authors carefully characterized with several characterization techniques such as morphological studies (FE-SEM), BET surface area, FT-IR, and also experimental studies which including modeling data, these scientific data’s are well organized and explained with sufficient manner, therefore I would consider for publication after the following comments have been addressed.
(i) On page No 8, authors have described that electrostatic interaction would be most possible based on their experimental studies from ionic species distribution (Figure.3a), I would appreciate from this journal reader perspective, it could provide more suitable experimental study, based on zeta potential of the as-prepared adsorbent, before and after modification would be highly essential in order to understand surface charges, and also active functional groups (active sites) would be available for adsorption studies of vanadium (V), this approach would be highly anticipated for adsorption phenomena.
(ii) Furthermore, after adsorption of the as proposed adsorbent, how efficient for regeneration studies carried over, how many cycles probably achieved, please provide these details this would be more reliable in environmental greener perspective, in order to support this phenomenon, author should provide FT-IR, Powder XRD before and after regeneration following necessary explanation would be discussed in the revised manuscript.
(iii) On page No. 9, 5 authors should provide more examples of SiO2, SiPAR, and resin-based adsorbents compared with in terms of adsorption capacity from the latest existing literature studies that would be itemized.
Author Response
Reviewer #3:
This study described a Removal of V(V) from solution using a silica-supported primary amine resin. The authors carefully characterized with several characterization techniques such as morphological studies (FE-SEM), BET surface area, FT-IR, and also experimental studies which including modeling data, these scientific data’s are well organized and explained with sufficient manner, therefore I would consider for publication after the following comments have been addressed.
(i) On page No 8, authors have described that electrostatic interaction would be most possible based on their experimental studies from ionic species distribution (Figure.3a), I would appreciate from this journal reader perspective, it could provide more suitable experimental study, based on zeta potential of the as-prepared adsorbent, before and after modification would be highly essential in order to understand surface charges, and also active functional groups (active sites) would be available for adsorption studies of vanadium (V), this approach would be highly anticipated for adsorption phenomena.
Author’s reply: Thank Reviewer so much for the valuable comments and the very detailed suggestions. Firstly, I am very regretful that experiments about zeta potential were not conducted in this work. But given that V(V) ionic species distribution and the protonation process of primary amine, it is enough to explain the experimental phenomenon. We has revised the related expression as “poor interaction between the protonated amino groups and VO2+ weakened the adsorption efficiency when the pH was lower than 4”. From the perspective of the whole paper, we think there is insignificant effect on this study without the analyses of zeta potential. We wish to thank Reviewer 3 again for having undertaken review of our manuscript and giving the constructive comments.
(ii) Furthermore, after adsorption of the as proposed adsorbent, how efficient for regeneration studies carried over, how many cycles probably achieved, please provide these details this would be more reliable in environmental greener perspective, in order to support this phenomenon, author should provide FT-IR, Powder XRD before and after regeneration following necessary explanation would be discussed in the revised manuscript.
Author’s reply: Thank Reviewer very much for the valuable comments. We apologize for that the regeneration of resins are not conducted in this study. The organic component of this silica-supported resin is essentially an anion exchange resin, with the same structure and performance as the traditional anion exchange resin, which can be reused many times. So we think the research about regeneration seems less necessary. Furthermore, this research aims at the experimental data combined with mathematical models. We find the SiPAR possesses faster kinetic process than the D302 resin, and further demonstrate this result by mathematical modeling.
(iii) On page No. 9, 5 authors should provide more examples of SiO2, SiPAR, and resin-based adsorbents compared with in terms of adsorption capacity from the latest existing literature studies that would be itemized.
Author’s reply: We thank Reviewer very much for the kind suggestions and valuable comments. Following the reviewer’s suggestions, we had made necessary corrections by adding more examples of other adsorbents from the recently published papers for the comparison in the Table 5.
[Page 17, line 564]: “……Table 5……”
Table 5
Comparison of sorption properties with other adsorbents for V(V).
|
Adsorbent |
C0 (mg/L) |
Adsorption capacity |
Equilibrium time |
Reference |
|
MZ-PPY |
25–250 |
65 mg/g |
About 90 mina |
[7] |
|
Zr(IV)-SOW |
- |
51.1 mg/g |
20 h |
[18] |
|
Cell-AE |
25–600 |
197.75 mg/g |
1 h |
[25] |
|
PGTFS–NH3+Cl− |
10–300 |
45.86 mg/g |
4 h |
[26] |
|
ZrO2 |
- |
54.3 mg/g |
24 h |
[34] |
|
Waste metal sludge |
7.6–48.4 |
24.8 mg/g |
7 h |
[35] |
|
CCSB |
20–150 |
148.15 mg/g |
9 h |
[36] |
|
GTMAC |
10-591 |
34.3 mg/g |
24 h |
[37] |
|
TiO2 nanoparticles |
3-800 |
50 mg/g |
30 min |
[38] |
|
D302 |
700 |
302.4 mg/g |
>3 h |
This work |
|
SiPAR |
50–400 |
70.57 mg/g |
90 min |
This work |
Notes: “-” means that the property was not provided; “a” denotes that the value was observed from the kinetics figure.
Round 2
Reviewer 1 Report
The TGA and acid-base titration have been carried out improving the quality of the work. Some interpretation of the TGA data is erroneous. In particular, the minor loss of weigth at 70-200C is not any kind of decomposition, but loss of adsorbed water/solvent or adsorbed CO2. The decomposition starts apparently at just below 300C. The content of -NH2 determned by titration (0.66 meq/g) is fully reasonable (but would correspond on decomposition to much more than 0.46%). The peaks at about 2300 cm-1 in the FTIR are not explained. They should be corresponding to adsorbed CO2 or formed carbonate. This needs to be commented in the manuscript.
Author Response
Author reply: Thank Reviewer very much for the valuable comments. Following the suggestion of the reviewer, the related description of TG and FT-IR in the has been clarified in the revised manuscript. [Page 7, line 238-239]: “…… Furthermore, the peak at 2360 cm-1 may result from the adsorbed CO2 in the samples ……”. [Page 7, line 250-253]: “…… For the SiPAR, there was a endothermic effect in the range of 70–200 °C, corresponded to the mass loss of 0.46%, which resulted from the loss of adsorbed water; a weight loss about 1% at 200–300 °C was due to the presence of primary amine functional groups ..….”.